

# Multiangle perception-oriented environmental facility design method based on joint fuzzy decision-making and transfer learning

Siconghui Yao

College of Art and Design, Sias University, Zhengzhou, China

## ABSTRACT

In modern society, the demand for environmental facilities is increasing, and how to effectively design and plan environmental facilities has become an urgent issue. However, traditional design methods often consider only certain requirements and perspectives, resulting in design results deviating from the expectations of actual users. In this study, first, perceptual fuzzy decision-making and design transfer learning were selected as methods. Second, by applying multiple perspectives to environmental facility design methods, these two methods were combined, and a new joint algorithm was proposed. Third, when designing environmental facilities, a joint processing framework was constructed considering the impact of human factors, environmental parameters, and cultural value parameters on the design results. Last, the proposed joint algorithm was validated for functionality and satisfaction. The experimental results of this article indicate that in temperature control design, the accuracy of this research model is 17.7–19.6% greater than that of traditional centralized algorithms. In terms of lighting design, the model results of this study are good, with an increase of 16.7–20.2%. This method comprehensively considers the various dimensional requirements of environmental facilities and has good migration performance. In future studies, we will further investigate the applicability of this method in different scenarios and applications to promote the further development of environmental design.

## INTRODUCTION

In recent years, with advancements in science and technology and social progress, people's requirements for environmental facility design have increased (*Dai & Meng, 2022*). As an emerging research field, environmental facility design methods oriented towards multiperspective perception are receiving increasing amounts of attention from both academia and industry (*Liu, Li & Liu, 2021*). This article focuses on the theme of the "Environmental Facility Design Method for Multiperspective Perception: Joint Fuzzy Decision and Transfer Learning Method", which is aimed at exploring a new method that comprehensively considers multiple factors in the process of environmental facility design

Corresponding author
Siconghui Yao, yaodh1990@163.com

(*Wunder, Engel & Pagiola, 2008*; *Novais, 2021*). Environmental facility design is the process of providing a place for human habitation, work, and life. The traditional environmental facility design process focuses on meeting basic functional requirements, such as safety, availability, and durability. However, with the increasing pursuit of quality of life and environmental awareness, designers are beginning to realize that more detailed factors such as comfort, sustainability, aesthetics, and user satisfaction need to be considered in design.

In this context, the design method of environmental facilities for multiangle perception has been proposed and has attracted widespread attention (*Ruder, 2019*). This method is aimed at achieving the optimal design of environmental facilities by comprehensively considering various factors and the needs of stakeholders. However, due to the complexity and diversity of the environmental facility design process, traditional design methods have certain limitations (*Dweiri & Meier, 1996*). To overcome these limitations and improve the effectiveness of design, this article proposes a joint fuzzy decision-making and transfer learning method to better address multiple perspectives on environmental facility design problems (*Hsueh, Sun & Zhang, 2022*).

The research objective of this article is to explore how to improve the comprehensive performance of environmental facility design by integrating fuzzy decision-making and transfer learning methods. Fuzzy decision-making methods can handle fuzzy and uncertain problems in design and provide decision-making results from multiple perspectives. The transfer learning method can utilize existing knowledge and experience to help solve problems in new fields. This article combines these two methods to achieve the goal of comprehensively considering multiple factors involved in environmental facility design. The research framework is shown in Fig. 1.

The main contributions of the study are listed as follows:

(1) A new algorithm combining fuzzy decision-making and transfer learning is established. This algorithm uses more data cores, which is more consistent with modern environmental facility design.

(2) This study jointly considers human factors, environmental factors, and cultural value factors, expanding the input content of joint calculation methods and enabling more refined treatment of environmental facility design.

The structure of the article is described as follows: The first part of the study elaborates on the necessity of this research. The second part provides an explanation of the current research progress. In the third part, two methods—perceptual fuzzy decision-making and design transfer learning—are selected and integrated based on the multiangle environmental facility design method. A new joint algorithm is proposed. In the fourth part, based on the design requirements of environmental facilities, human factors, environmental parameters, and cultural value parameters are selected as factors of the design results, and a joint processing framework is established. The final section conducts functional and satisfaction verification on the proposed joint algorithm.

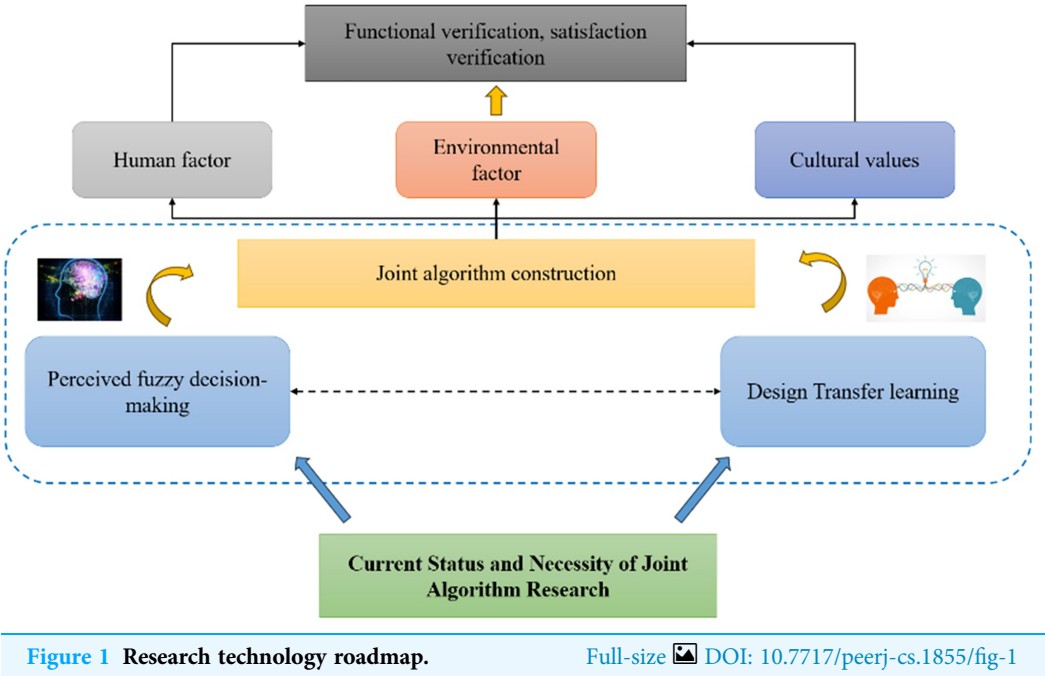

**Figure 1** **Research technology roadmap.** 

# LITERATURE REVIEW

The acceleration of urbanization worldwide and the design of environmental facilities are crucial for improving the quality of life of residents and the sustainable development of cities. Traditional design methods are based mainly on expert experience and static data, making it difficult to meet the design requirements of different scenarios and requirements. Therefore, this article proposes a design method for environmental facilities based on multiangle perception, which extracts effective design strategies from multidimensional data through joint fuzzy decision-making and transfer learning methods. With the rapid development of digitalization and intelligent technology, environmental facility design methods based on multiangle perception have become increasingly important, and an increasing number of scholars have conducted related research (*Tirkolaee et al., 2020*).

*Xu et al. (2022)* focused their research on the application of environmental perception and decision-making methods. The authors explored environmental perception methods based on fuzzy logic and multimodal data and combined transfer learning to improve model performance. They modelled various factors using fuzzy set theory and applied transfer learning methods to transfer existing knowledge to a new environment, improving the accuracy and reliability of design solutions. *Wen et al. (2021)* focused on urban facility design and sustainable development using fuzzy decision models and transfer learning methods to evaluate and optimize the design of environmental facilities in urban planning. The study by *Zhu (2022)* focused on multiangle perception and decision-making methods in building environments. Using fuzzy logic and transfer learning techniques, an environmental facility optimization method was designed to provide better indoor comfort

and energy efficiency. *Asokan et al. (2020)* combined fuzzy decision-making and transfer learning techniques to improve design effectiveness and decision performance using a multiperspective perception approach to environmental facility design. Their research is aimed at optimizing environmental facilities to meet different user needs and provide a better user experience. *Maurya et al. (2021)* have extensive research interests in urban research and sustainable development. Her research team explored the application of joint fuzzy decision-making and transfer learning methods in urban environmental facility design and proposed a design program evaluation and optimization method based on multiangle perception by considering various factors. *Nawaz & Gul (2022)* focused on the application of transfer learning in environmental facility design. The authors explore how to use transfer learning technology to migrate existing knowledge and data migration to a new environment to improve the effectiveness and feasibility of the design scheme. *Huang et al. (2022)* promoted advanced cognitive functions such as cross-modal integration, recognition, and imagination *via* integration and interaction in human multisensory networks to accurately evaluate and comprehensively understand the multimodal world. The authors proposed a new cross-modal integration framework for upper limb prosthetics based on the type 2 fuzzy logic system (FLS).

The research of these scholars has provided an important theoretical basis and practical guidance for environmental facility design methods oriented towards multiperspective perception and has made significant contributions in related fields. With continuous related research, more scholars, including our team, have joined this research field and proposed more innovative and effective methods.

## FUNDAMENTALS OF RESEARCH THEORY

### Perceived fuzzy decision-making

In a decision-making environment, due to complexity and cognitive limitations, some decision information that cannot be represented by real numbers may be encountered (*Bukhari, Tusseyeva & Kim, 2013*). To describe this uncertain qualitative information, people can use fuzzy information, such as the Hierarchical Fuzzy System (HFS), Hesitant Intuitionistic Fuzzy Linguistic Term Set (HIFLTS), or probabilistic linguistic term set (PLTS). Each perceptual variable in the HFLTS has equal importance and requires continuous perceptual expression, which slightly limits the expression of individual or group opinions. A PLTS assigns corresponding probabilities or weights to each language term in the set to distinguish the importance of different language terms, thereby better representing uncertain information in decision-maker cognition (*Zhang et al., 2022*). As an extension of the HFLTS, the PLTS provides more refined language terminology expression ability and better reflects the degree of importance that decision-makers attach to each language term. Each language term in the PLTS has a corresponding probability or weight, so that the proportion of each language term in decision-making can be calculated based on probability or weight (*Arthington et al., 2010*). Moreover, the PLTS can effectively integrate the subjective opinions of multiple DMs, and the information contained in these subjective opinions can be uniformly characterized and described through the PLTS method.

**Peer**J Computer Science

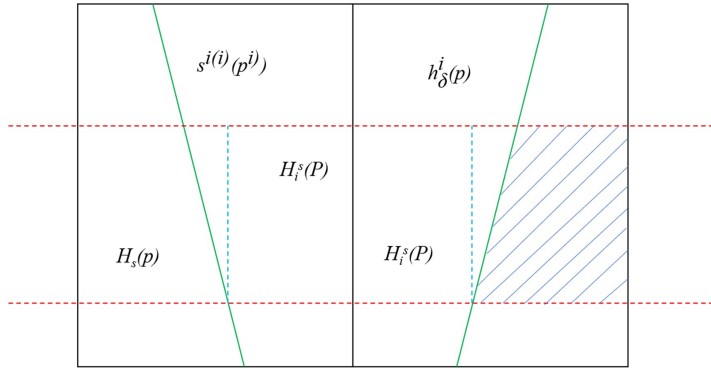

**Figure 2  Schematic diagram of the modified containment set.**

Let $x_i \in X$ be a fixed set and $S = \{s_{-\tau} \ldots, s_{-1}, s_0, s_1 \ldots, s_\tau\}$ be a set of perceptual meanings; the PLTS on $S$ is defined as *Huang, Keisler & Linkov (2011)*:

$$H_s(p) = \{ < x_i, \ h_s^i(p) > /x_i \in X\} \tag{1}$$

$$h_\delta^i(p) = \{s^{i(i)}(p^i)) \mid s^{i(i)} \in S, \ p^{(i)} \geq 0, \ \ l = 1, 2 \ldots, L, p^{(i)} \leq 1\} \tag{2}$$

Here, $s^{i(l)}(p^{(l)})$ is the language term $s^{i(l)}$ in the PLTS, where the probability is $p^{(l)}$. $h_s^i(p)$ is referred to as the probability hesitant fuzzy language element that represents the number of different language terms in $h_s^i(p)$.

Because the number of fuzzy perception meanings in the PLTS may be different, certain difficulties may arise in the operations between two PLTSs. To facilitate the operation and comparison of PLTSs with different numbers of g perceptual terms, an extended method is proposed: for any two $h_s^1(p)$ and $h_s^2(P)$, if $L_1 > L_2$, fuzzy perceptual meanings can be added to $h_s^2(P)$, where the probability of the added fuzzy perceptual meanings is zero. This expansion method does not change any original information about the PLTS, effectively preventing information loss. We refer to this as the expanded PLTS standardization, and the modified PLTS set has stronger fuzzy scale inclusivity. The results are shown in Fig. 2.

Related studies have shown that the psychological behaviour of decision-makers manifests as risk aversion towards losses and risk preference towards returns. To model the psychological behaviour of decision-makers in uncertain environments, prospect theory studies people's psychological behaviour in risky decision-making situations according to the following basic principles:

(1) Reference dependency: The result of the value function is based on the gains and losses of a reference point.

(2) Decreasing sensitivity: The risk attitude of individual decision-makers manifests as a preference for return risk and risk aversion for loss.

(3) Loss aversion: Individuals are more sensitive to losses than to gains. Based on the above principles, the value function of prospect theory is defined as the deviation function from the reference point, and the gains and losses are defined as the concave S-shaped function and convex S-shaped function, respectively.

To characterize the sensitivity of decision-makers to gains and losses, the value function of prospect theory is defined as follows (*Kaya, Karaşan & Güvercin, 2023*):

$$v(x) = \begin{cases} x^{\alpha}, & x \geq 0 \\ -\lambda(-x)^{\beta}, & x < 0 \end{cases} \tag{3}$$

where $x$ represents profit ($x \geq 0$) or loss ($x < 0$) and $\alpha(\alpha \in [0, 1])$ and $\beta(\beta \in [0, 1])$ are two exponential parameters. $\lambda(\lambda \geq 1)$ is the individual's level of risk avoidance. $\lambda$ represents that the greater the value is, the stronger the individual's risk aversion attitude, as verified in *Bao, Zhang & Zhou (2022)*.

## Design transfer learning

Design transfer learning involves borrowing knowledge and experience from existing facility design to transfer it to new environmental design issues and provide better solutions. Transfer learning can provide guidance and inspiration for new environmental design problems by learning and utilizing features, rules, or experiences from existing design cases. The fuzzy decision-making method can handle the uncertainty and complexity of fuzzy thinking and problem decision-making and assist in balancing the commonality and individuality of environmental facility design through the modelling and analysis of fuzzy logic and fuzzy mathematics. For the distributions of $X_s$ and $X_t$ in the source domain data sample and daily standard domain data sample, which are $P(X_s)$ and $P(X_t)$, respectively, the distance measured using the MMD is *El Bourakadi, Yahyaouy & Boumhidi (2022)*:

$$MMD(P_s, P_i) = \left\| \frac{1}{n_s} \sum_{i=1}^{n} \phi(x_{s_i}) - \frac{1}{n_i} \sum_{j=1}^{n} \phi(x_g) \right\| \tag{4}$$

where $\phi(x)$ represents infinite order nonlinear feature maps in Hilbert space; $X_s = \{x_{si}\} = \{\phi(x_{si})\}$ and $X_t = \{x_{ii}\} = \{\phi(x_{ii})\}$ represent the mapped data of the source domain and target domains, respectively; and ns and nt represent the number of samples in the source domain and target domains, respectively. Therefore, the difference in the data distributions in the source and target domains can be measured with Eq. (4).

However, from a probability distribution perspective, Eq. (4) considers only the edge probability distribution of the data sample. For oscillation risk identification problems, the cross domain is obtained from the data samples and label samples of two systems, and it is difficult to ensure that the classification boundary of the cross domain is consistent with the target domain by considering only the data samples. Therefore, in the computer field, the difference measurement algorithm with joint probability distribution adaptation (JDA) for classification problems also considers the label sample difference U39, and its learning process is shown in Fig. 3 below.

Through the joint distribution adaptation algorithm, we can fully consider the differences between the target domain and the source domain while utilizing the information of labelled samples to more accurately characterize classification boundaries. This approach is beneficial for addressing the challenges in designing environmental facilities from multiple perspectives. From a probability distribution perspective, our method (Eq. (4)) considers only the edge probability distribution of the data samples.

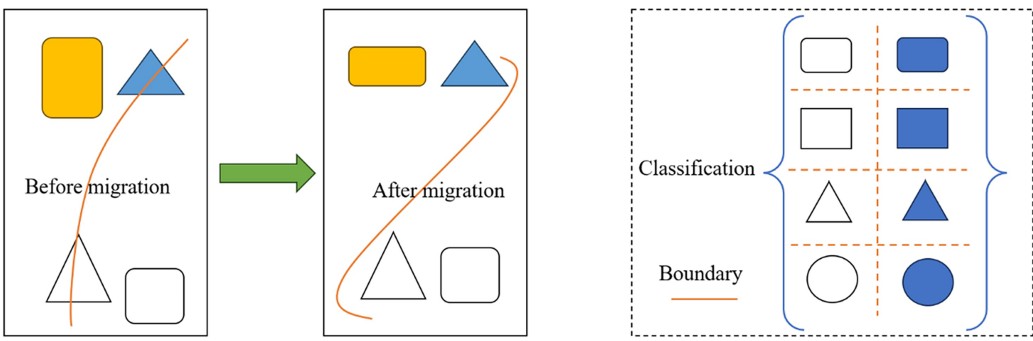

**Figure 3 Schematic diagram of migration adaptive learning process.**

In practical situations, the cross domain of the oscillation risk identification problem is obtained through data samples and label samples from two systems. It is difficult to ensure that the classification boundaries of the cross domain are consistent with those of the target domain by considering only the data samples. To solve this problem, in the field of computer science, a joint distribution adaptation (JDA) difference measurement algorithm, which simultaneously considers the differences in label samples, has been proposed for classification problems. The learning process of this algorithm is shown in Fig. 2. Our research method combines techniques such as the joint distribution adaptation algorithm, fuzzy decision-making, and transfer learning to solve the problem of multiperspective perception in environmental facility design. By jointly considering the differences between data samples and label samples and utilizing the idea of transfer learning, our method can provide more accurate and comprehensive design solutions, thereby improving the effectiveness of environmental facility design and the work efficiency of designers. The calculation formula is *Prakash & Barua (2015)*:

$$MMD(Q_s^{(c)}, Q_i^{(c)}) = \left\| \frac{1}{n_i^i} \sum_{i=1}^{n_i^i} \phi(x_n^i) - \frac{1}{n_i^i} \sum_{i=1}^{n_i^i} \phi(x_n^i) \right\| \tag{5}$$

where $c$ represents a label in category space $C$; $x_{si}^c$ and $x_{ti}^c$ represent samples labelled with c in the data domains $X_s$ and $X_t$, respectively. In the source domain, $y(x_{si}) = c$ represents the true label of the sample, while in the target domain, prelabelled labels are employed.

The regularization formula of the joint probability distribution adaptation algorithm can be obtained by simultaneously solving Formulas (4) and (5):

$$D_H = MMD(P_s, P_t) + \sum_{c=1}^{C} MMD(Q_s^{(c)}, Q_t^{(c)}) \tag{6}$$

By adding the regularization term shown in Formulas (3)–(6) to the initial CNN data-driven model, a general CNN learning framework based on the joint probability distribution adaptation algorithm is obtained. The learning objectives are *Yang, Zhang & Tian (2023)*:

$$\min_{f_{cxx} \in H} \left[ \sum_{k=1}^{s} L(f_{cNW}(\Delta x_k), y_k) + \sigma \|f_{cnw}\| + \gamma D_H \right] \tag{7}$$

where γ represents a regularization system and σ is a network parameter.

## Joint algorithm

When designing joint algorithms, the parameters and structure of the algorithm need to be adjusted based on specific problem scenarios and data characteristics to better adapt to current needs. This personalized optimization process helps improve the performance of the algorithm and better adapts it to practical applications. Designing a joint algorithm for fuzzy decision-making and transfer learning requires customization and optimization based on specific application scenarios and data characteristics. The advantage of this method is that it combines the advantages of two different learning and decision-making methods, thereby improving the robustness and accuracy of classification and decision-making. However, factors such as algorithm complexity, computational resources, and data availability must be considered.

Thus far, through the minimization of Formula (7), the marginal distributions of the source domain and target domain and the moment statistics of the conditional distribution of each order can be adapted in the Hilbert space. However, according to Eq. (3), we need to solve the kernel mapping when calculating the joint probability distribution regular term D φ, which may lead to an infinite dimensional space. Therefore, to facilitate calculation, a kernel function is introduced, and Eq. (7) is rewritten in kernel form *Prakash & Barua (2015)*:

$$D_H = tr(\beta^7 \boldsymbol{KLK}\beta) + \sum_{c=1}^{C} tr(\beta^7 \boldsymbol{KL_cK}\beta) \tag{8}$$

$\beta$ are the model parameters that represent the composition of network parameters; that is, $f_{CNN}$ is represented as follows:

$$\begin{aligned} f_{CNN} &= \sum_{i=1}^{s} \beta_i K(x_i, x) \\ &= \sum_{i=1}^{S} \beta_i \phi(x_i)\phi(x) \\ &= \omega^T \phi(x) \end{aligned} \tag{9}$$

In Eq. (9), $K$ represents the relationship between $\phi(x)$ and the corresponding kernel function, with $K(x_i, x) = <\phi(x_i), \phi(x)>$.

In real-world environmental facility design applications, selecting suitable machine learning models and evaluating their performance are crucial. The selection of models is closely related to the accuracy and error rates. In classification problems, we usually use the error rate and accuracy to evaluate the performance of the model. The error rate refers to the proportion of misclassified samples to the total number of samples included in the evaluation, while the accuracy rate refers to the proportion of correctly classified samples to the total number of samples included in the evaluation. The accuracy can also increase with the accuracy of the model, which represents the accuracy of the model in classification problems. In real-world environmental facility design applications, multiple rounds of training are usually necessary to continuously update the parameters of the machine learning model. During the training process, selecting the best model is crucial for determining its performance in practical applications. Therefore, when evaluating a model, two aspects need to be considered, namely, selecting the appropriate model during the

**Table 1 Multi angle joint transfer learning settings.**

| Data sources | Migration task | Source domain | Target domain | Number of styles | Random state |
|---|---|---|---|---|---|
| User questionnaire survey | Intranuclear migration | 1,000 | 1,800 | 10,000 | Fuzzy transfer learning centered on satisfactory design |
| | Fuzzy transfer | 1,600 | 2,400 | 10,000 | |
| | Extranuclear migration | 2,200 | 3,000 | 10,000 | |

training process and conducting performance testing on the saved model. For model evaluation in real-world facility design applications, we need to focus on two aspects: model selection and performance testing. Choosing the best model and ensuring its accuracy are key factors in ensuring the performance of the model. By accurately evaluating the accuracy and error rate of the model, we can better understand its performance and make corresponding adjustments and improvements (*Chen & Duh, 2008*). The classification accuracy is the accuracy of the model. In Formula (9), the kernel matrix $K$ is composed of the kernel matrix in the source domain, target domain and cross domain; L is the semip-positive-definite matrix corresponding to the marginal distribution; and $L_c$ is the semip-positive-definite matrix corresponding to the conditional distribution, which are expressed as *Honarvar & Sami (2019)*:

$$K = \begin{bmatrix} K_{s.s} & K_{s.t} \\ K_{t.s} & K_{t.t} \end{bmatrix} \tag{10}$$

$$L_\psi = \begin{cases} \frac{1}{n_i^2} & x_i, \ x_i \in X \\ \frac{1}{n_i^2} & x_i, \ x_j \in X \\ -\frac{1}{n_i n_i} & \equiv lim_i lim_i \to lim_i \to \infty \end{cases} \tag{11}$$

Eqs. (10) and (11) represent the frameworks of the transfer learning algorithm based on the CNN. According to the training process of the CNN model, backpropagation is used to update the model parameters, and the fCNN is the oscillation risk identification model. Notably, reducing the difference between the source domain and the target domain is the key to achieving effective application of the oscillation risk identification model (*Zhao et al., 2021*; *Prabakaran, Vaithiyanathan & Ganesan, 2022*). This section only uses typical difference measurement algorithms as examples. For different engineering problems, multiple methods can be employed to reduce the feature differences between two similar systems. The specific multiangle joint algorithm learning settings are shown in Table 1.

To avoid overfitting during model training, it is necessary to evaluate the model during the training process. Cross-validation is the main method of model evaluation. The principle of cross-validation is to group the original training samples into a training set and a validation set and calculate the training and validation sets based on the indicator classification of Eq. (11) in one training cycle. The training set is used for classifier training (calculating loss and backpropagation). The purpose of the validation set is to evaluate the current model and select the model.

## MULTIANGLE ENVIRONMENTAL FACILITY DESIGN

Human perception is a crucial aspect of design, as environmental facilities should be able to meet people's needs and provide a comfortable user experience. Designers need to consider people's perceptions of spatial layout, colour, material texture, and other aspects and combine ergonomic principles to ensure that users feel comfortable and satisfied when using these facilities. Social needs comprise another perspective that needs to be considered. Designers should understand and consider the role of environmental facilities in society and their impact on various populations. For example, the design of public transportation facilities should take into account the needs of elderly people, disabled people, and children to ensure that they can easily use these facilities. Sustainability is another key perspective. The design of environmental facilities should focus on resource conservation and environmental protection. For example, in architectural design, renewable materials, efficient energy systems, and water recycling systems can be utilized to reduce their impact on the environment. The multiperspective environmental facility design method is aimed at creating a comfortable and sustainable environment that meets user needs by comprehensively considering various aspects, such as human perception, social needs, sustainability, and safety.

### Human parameter selection

In the design of multiangle environmental facilities, the selection of human parameters plays an important role. The artificial parameters refer to adjustable parameters set by designers during the design process based on specific needs and goals to achieve the best design effect. The goal of human parameter selection is to find the most suitable design scheme for specific environmental facilities by comprehensively considering different perspectives and factors (*Xu et al., 2018*). Designers should reasonably select and set human parameters based on the background and objectives of the project, as well as the needs and preferences of users. The selection of human parameters should also be combined with expert opinions and user feedback. Designers can collaborate with experts in relevant fields and use their domain knowledge and experience to guide parameter selection (*Zappone, Di Renzo & Debbah, 2019*). Moreover, designers should collect and analyse user feedback and need to understand user preferences and expectations for design parameters to adjust and optimize parameter selection.

The membership function of facility user satisfaction is $\mu_{Bi}$, as shown as follows.

$$\mu_{Bi}(N) = \begin{cases} 1 & 0 \le N \le 23/3 \\ -0.6N/B + 1.2 & 2B/3 < N \le 2B/3 \\ -2.4N/B + 2.4 & 2B/3 < N \le B \\ 0 & B/B \end{cases} \tag{12}$$

$$\mu_{n,B}(N,n) = \begin{cases} N/n & 0 \le N \le min(n,B) \\ 1 & n < N \le B \\ 0 & \notin N \end{cases}$$

where N represents the number of people in the facility at the same time and B represents the maximum amount of time.

By considering various user preferences and needs, environmental facilities that are more consistent with user expectations can be designed, further improving user satisfaction and experience. Improving environmental adaptability: The reasonable selection of human factor parameters can increase the adaptability of environmental facilities, enabling them to meet the needs and perceptions of different users and improving the adaptability and universality of environmental facilities. Optimizing resource utilization: By considering human factor parameters, environmental resources can be better planned and utilized, improving resource utilization efficiency and achieving maximum utilization and conservation of resources.

## Environmental parameter selection

Environmental parameters refer to parameters related to a specific environment, including but not limited to climate, geographical conditions, air quality, and lighting conditions. In the selection of environmental parameters, designers need to comprehensively consider the impact of various environmental factors on the design scheme and the needs of users. Each region has unique geographical characteristics, and corresponding environmental parameters need to be selected based on the actual situation. The indoor and outdoor air quality directly impacts people's health and comfort. Therefore, in the design of environmental facilities, factors such as ventilation systems and air purification devices must be considered to ensure good air quality. The reasonable use of natural light can improve the comfort and energy efficiency of buildings. Designers need to select appropriate environmental parameters based on factors such as the lighting angle, building orientation, and surrounding environmental characteristics to maximize the use of natural light.

The environmental parameters describe the overall similarity of the data, the conditional distribution describes the similarity of each category, and the joint probability distribution is formally obtained by adding the marginal distribution and conditional distribution. This research assumes that the fields participating in learning obey the same conditional probability distribution, approximate the marginal distribution distance to the joint probability distribution distance between two fields, and realize marginal distribution adaptation *via* the distance measurement method based on the HFLTS to reduce data distribution differences and realize transfer learning.

First, the Gaussian kernel is selected as the kernel function in an HFLTS, which is mathematically represented as *Kale et al. (2015)*:

$$k(x_i, x_j) = exp\left( -\frac{\| x_i - x_j \|^2}{2\sigma^2} \right) \tag{14}$$

In the equation, $\sigma$ is the width of the kernel function.

Second, by calculating the difference in the distributions of the data features between the source domain and the target domain corresponding to the domain adaptation layer, the distance metric between two domains is obtained as follows:

$$HFLTS(P(F'_S), P(F'_T)) = \left\lVert \frac{1}{n_S} \sum\nolimits_{i=1}^{n_S} \phi(F_S) - \frac{1}{n_T} \sum\nolimits_{i=1}^{n_T} \phi(F_T) \right\rVert^2 \tag{15}$$

In the formula, $F$ is the output vector of the first layer of the neural network and $\phi(\cdot)$ is a nonlinear mapping function in an HFLTS.

This chapter uses the sum of the higher-order moment matching function and second-order moment matching function as the domain adaptation loss and adds both terms to the regularization term of the objective function of the deep transfer network, thereby achieving joint matching of feature distribution moments in the source and target domains.

### Selection of cultural value parameters

This article proposes a multiperspective perception-oriented environment facility design method that combines two machine learning methods, namely, fuzzy decision-making and transfer learning. The selection of cultural value parameters considers the impact of specific cultural environments and cultural differences on the design scheme in the design of multiangle environmental facilities. Cultural value parameters include but are not limited to cultural relic protection, cultural inheritance, cultural exchange, or other aspects. In the process of selecting cultural value parameters, designers need to understand and consider the characteristics of different cultural environments and backgrounds to provide environmental facility design solutions that meet local cultural needs and values. Moreover, designers need to consider the impact and influence of various cultural value factors on environmental facility design from a professional and interdisciplinary perspective, combined with historical, social, cultural and other factors. We suggest selecting cultural value parameters from the following aspects:

(1) Cultural symbols and symbols: Different cultures have different symbols and meanings. By investigating and understanding the symbols and symbols of local culture, relevant cultural value parameters are selected.

(2) Social interaction norms: Different cultures have different norms and values for social interaction and etiquette. These norms and values are considered in the design, and the corresponding cultural value parameters are selected.

(3) Aesthetics and aesthetic standards: Different cultures have different preferences and standards for aesthetics and aesthetic standards. Therefore, in the design, the aesthetic requirements and preferences of local culture are taken into account and considered cultural value parameters.

Reasonable selection of cultural value parameters enhances cultural adaptability in environmental facility design. By considering local cultural values and customs, environmental facilities that are more consistent with local cultural backgrounds are designed, enabling users to better adapt and accept them. Improving user experience: Reasonable consideration of cultural value parameters can enhance user comfort and satisfaction and enhance user experience of environmental facilities. Promotion of sustainable development: By reasonably selecting cultural value parameters, environmental facilities that are consistent with local cultural and environmental

characteristics can be designed to promote sustainable development and reduce unnecessary environmental conflicts.

### Joint processing framework

Joint fuzzy decision-making and transfer learning provide powerful support for the design process of environmental facilities oriented towards multiangle perception. The joint processing framework established in this study is presented as follows:

(1) Data preparation stage: Relevant datasets are collected from the source and target domains, and features are preprocessed and extracted from the data for subsequent processing.

(2) Transfer learning stage: A pretrained model, such as gpt-3.5-turbo, serves as the benchmark model, the model is trained from the source domain dataset, a common feature representation is obtained, and the model is saved as a migration model for subsequent transfer learning.

(3) Fuzzy decision stage: Appropriate fuzzy logic methods, such as fuzzy neural networks or fuzzy inference systems are selected; appropriate input and output variables are selected and defined; fuzzy rule systems are designed; and fuzzy rules are trained and optimized according to training data to obtain accurate decision models.

(4) Joint processing stage: The migration model is loaded, the general feature representation of the source domain data is loaded, the general feature is input into the fuzzy decision model for decision-making, and the design parameters of environmental facilities are optimized according to the output of the fuzzy decision to meet the needs and requirements of multiangle perception.

(5) Evaluation and tuning: Test datasets are used to evaluate the performance and effectiveness of joint processing methods, and the results are tuned and improved based on the evaluation results to improve the accuracy and adaptability of the design.

Through the above steps, we can construct a simple joint processing framework to achieve a multiperspective perception-oriented environmental facility design method. Of course, in practical applications, further development and optimization of the framework may be needed according to the specific situation. The pseudocode is shown in Table 2.

## TEST AND VERIFICATION

To verify the functionality of the joint fuzzy decision and transfer learning algorithm in the multiperspective perception-oriented environmental facility design method, the following verification examples are designed to verify the functionality and satisfaction of the algorithm to provide a comfortable and adaptive environment for different groups of people. Based on previous work (*Kale et al., 2015*), we implement the algorithm code. We hope to make design decisions using joint algorithms, taking into account user preferences and cultural backgrounds. We collect user preference information and local cultural background information. For example, a survey questionnaire can be utilized to collect users' preferences for seat style, colour, material, and layout. Simultaneously, the traditional and aesthetic requirements of local culture for seat layouts were collected. Based on the collected user preferences and local cultural background, a joint algorithm model

**Table 2  Joint algorithm pseudocode.**

1:   Input: i, data training frequency $M_P$ and overall size n;

2:   for all iP= 1 to Mp do

3:     for all i = 1 to n do

4:         Randomly split based on collected data

5:         while tabu (n, i) = 0

6:             for L = 1 : n

                    Complete transfer kernel learning based on fuzzy decision-making

                end for

7:             If the data changes meet the design requirements

                    Caculate the ht based on Eq. (9)

                Else iP = 1

                end if

8:             If the kernel requirements for Transfer learning are not met

                Select data that can be calculated within other ranges of change

                else

                Select other warning data that can reflect the real situation

                end if

              end for

9:       Initialize the federated model kernel

10:     If is_ Kernel (predicted_value):

11:     Print ("Facilities {} is at design". format (i))

12:   end for

13:   Print ("Meets facility design requirements {}". format (i))

based on fuzzy decision-making and transfer learning is established. The input features of the training model include seat layout parameters, user preference labels, and cultural value parameters. Based on the output of the joint algorithm, which is the result of fuzzy decision-making, the design decision of the seat layout is made.

## Functional verification

The functional verification design is aimed at verifying the effectiveness and feasibility of the proposed environmental facility design method in practice. The experimental design clarified the functional verification indicators of the design method. First, we clarify the functional validation indicators of the design method, which should be relevant to the research topic and capable of quantitatively evaluating the performance of the design method in practical applications. In the validation experiment, necessary data are collected, including environmental perception information, user feedback, and running results. Based on the data analysis results, the performances of the design methods are
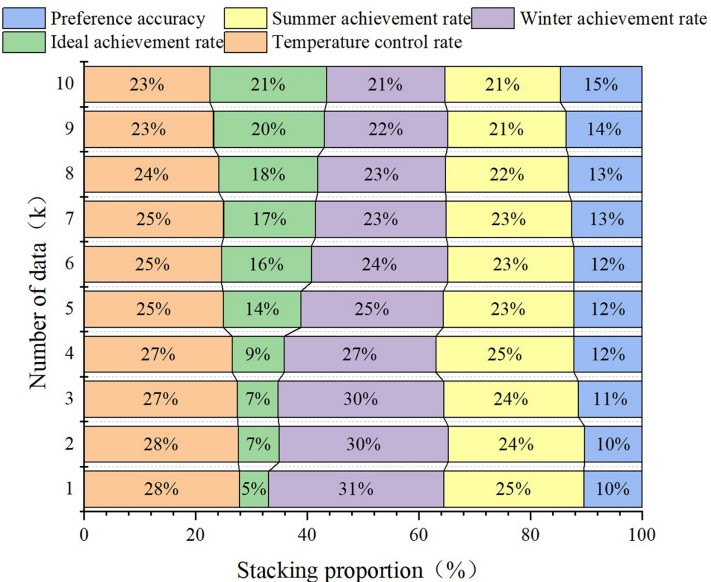

**Figure 4 Relationship between temperature control facility design and data.**

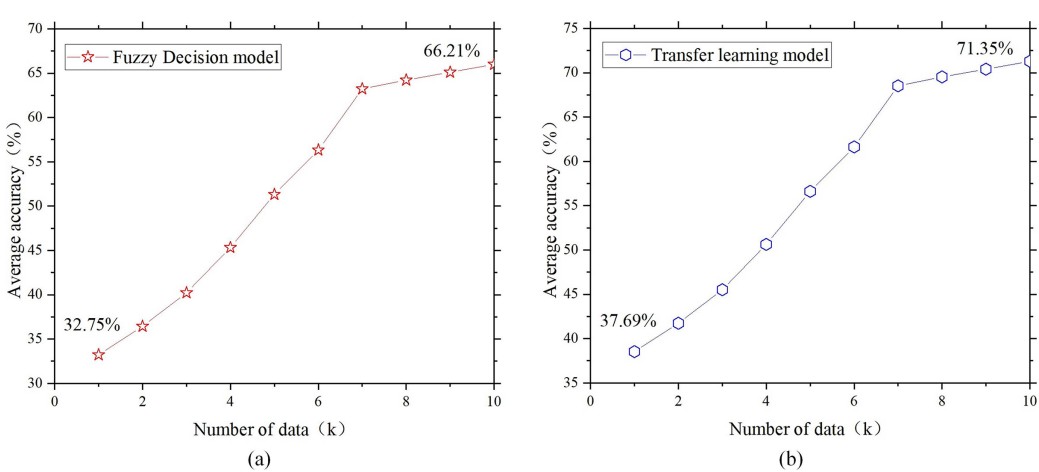

**Figure 5 Single model temperature control design.** (A) Fuzzy decision model; (B) transfer learning model.

evaluated and compared. The proposed method is compared with traditional or other related methods to evaluate its functional advantages. The results are shown in Figs. 4–6.

Figures 4–6 show that the accuracy of all the controls on the design of the temperature control facilities improves with increasing data volume, which is basically consistent with the actual situation. Because the data volume is increasing, the joint model can have more space for screening, grouping and transfer learning. Additionally, the "fuzziness" of fuzzy decision-making is decreasing, and the accuracy rate will naturally increase. However, some data remain basically unchanged after reaching a certain threshold. The temperature control rate was initially 53.23%, but as the data volume increased, it increased to approximately 86%. This is one of the most accurate starting points in temperature control

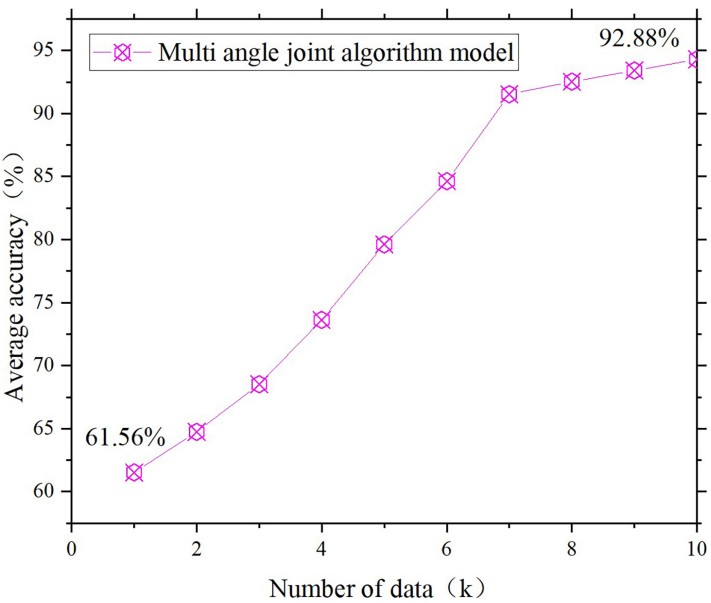

**Figure 6 Joint model temperature control design.**

design. However, when the data reserve reached 1K, the ideal achievement rate was only 10%, but it ultimately increased to 80%. This finding indicates that in the early stages of poor data quality, the joint model can continuously collect data, complete transfer learning, and improve its accuracy. The figure also compares the results of the single fuzzy decision model, transfer learning model and multiangle joint model on the design of the same environmental facilities. The results show that the accuracy of the multiangle joint model is between 17.7% and 19.6% greater than that of a single fuzzy model and transfer learning model. This finding directly indicates that the joint model established in this study can match the functionality of environmental facility design.

## Satisfaction verification

Our goal is to design a comfortable indoor office lighting system that meets the needs of multiple user groups. A joint algorithm is adopted to consider user preferences and accessibility requirements. First, user preferences and accessibility requirements, including lighting brightness, colour temperature, light source type, and lighting layout, are collected. Second, a joint algorithm model of fuzzy decision-making and transfer learning is trained on the use of lighting system parameters, user preference labels, and accessibility requirement parameters. During the training process, transfer learning methods are applied to transfer previous lighting design knowledge to new design tasks. Third, another set of data was utilized to evaluate the model to verify its impact on user satisfaction. The model is trained through lighting datasets, and the impact of design solutions on user satisfaction is evaluated through methods such as on-site evaluation, questionnaire surveys, or user experience testing. Last, we compare the user satisfaction of the joint algorithm and traditional design to verify the effectiveness of the joint algorithm in meeting the needs of multiple users. In this way, we can design an indoor office lighting

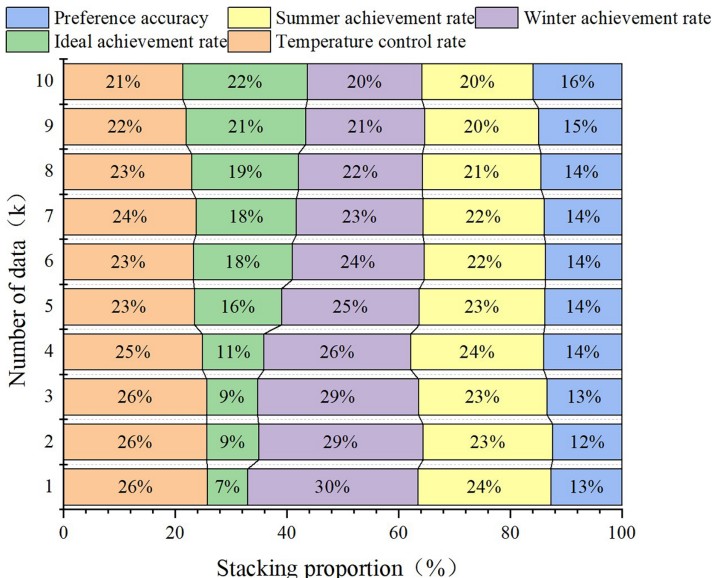

**Figure 7 Relationship between lighting facility design and data.**

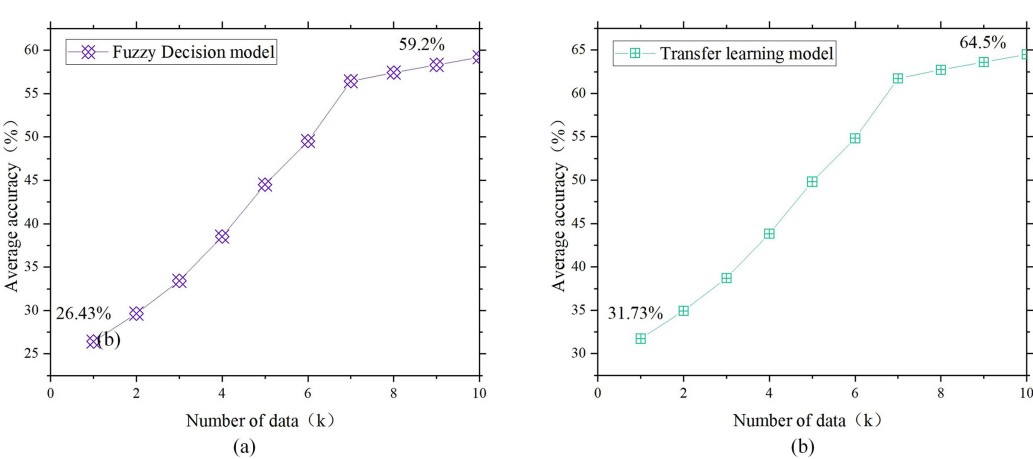

**Figure 8 Lighting design of single model.** (A) Fuzzy decision model; (B) transfer learning model.

system that is suitable for the needs of multiple user groups. The final results are shown in Figs. 7–9.

Figures 7–9 show that the validation results of the joint model on user satisfaction are basically consistent with the functional validation results. The satisfaction with lighting brightness increased from the initial value of 46.43% to approximately 79%, which is approximately 20% lower than the maximum value of functional validation. This situation is related to the higher requirements for lighting. Colour satisfaction was only 13% when the data reserve was 1K, but it ultimately reached 83%, which is similar to the ideal achievement rate in functional verification. This finding also indicates that the joint model can continuously improve its accuracy by collecting data and completing transfer learning

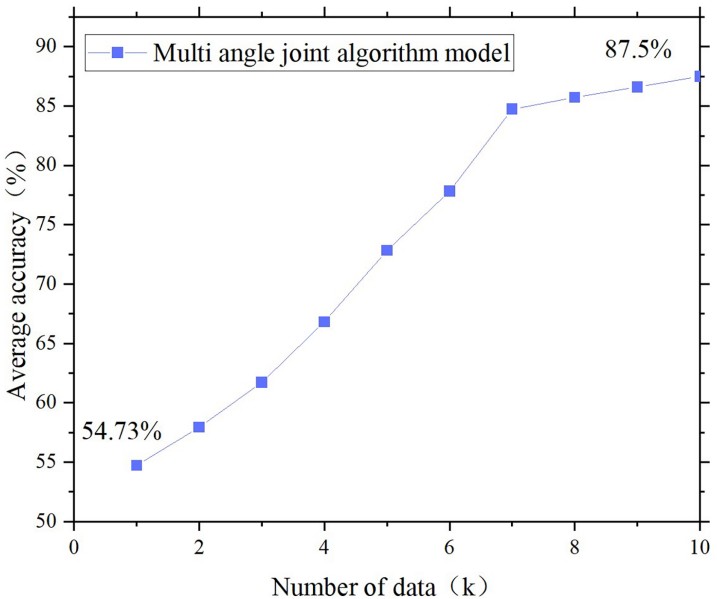

**Figure 9 Joint model lighting design.**

**Table 3 Method comparison summary.**

| Model name | Average accuracy | Achievement rate | Stability score |
| --- | --- | --- | --- |
| Ours | 86 | 93 | 2.35 |
| *Tirkolaee et al. (2020)* | 78 | 88 | 1.42 |
| *Maurya et al. (2021)* | 79 | 84 | 1.59 |
| *Arthington et al. (2010)* | 82 | 85 | 1.88 |
| *Huang, Keisler & Linkov (2011)* | 80 | 86 | 2.11 |
| *El Bourakadi, Yahyaouy & Boumhidi (2022)* | 81 | 85 | 1.59 |
| *Prakash & Barua (2015)* | 76 | 84 | 2.05 |
| *Prakash & Barua (2015)* | 81 | 82 | 2.13 |

in the early stage of poor data quality. The results of analysing the design of facilities in the same environment using fuzzy decision models, transfer learning, and multiangle joint models show that the accuracy of the multiangle joint model is between 16.7% and 20.2% greater than that of a single fuzzy model and transfer learning model, which is similar to the results of functional testing and consistent with the actual situation. The joint model has better performance.

A summary of the results is shown in Table 3.

Overall, the model proposed in this study performed the best, with an average accuracy of 86, a success rate of 93, and a stability score of 2.35. This finding indicates that our model has high prediction accuracy and robustness, making it suitable for various application scenarios. The performance of other models is relatively low, and specific application scenarios and requirements need to be considered when making choices.

## CONCLUSION

In this study, we synthesized joint fuzzy decision-making and transfer learning methods and proposed a design method for environmental facilities oriented towards multiangle perception. This method is aimed at assisting in comprehensive decision-making related to actual environmental facility design and achieving comprehensive analysis of environmental perception perspectives *via* joint fuzzy decision-making and transfer learning methods, thereby improving the efficiency and accuracy of environmental facility design. The study considered human, environmental, and cultural value parameters that affect the design results and constructed a joint processing framework. Functional and satisfaction verification were conducted on the proposed joint algorithm. The research results indicate that the joint model proposed in this study can complete transfer learning through continuous data collection in cases of poor initial data quality to improve its accuracy. In the design of temperature control, the accuracy of the multiangle joint model is 17.7% to 19.6% greater than that of the single fuzzy model and transfer learning model. In lighting design, the accuracy of the multiangle joint model is 16.7% to 20.2% greater than that of the single fuzzy model and transfer learning model. These findings indicate that the proposed joint model has several advantages. Moreover, transfer learning can continuously improve the accuracy in situations of poor data quality and achieve higher accuracy in temperature control and lighting design than traditional fuzzy models and transfer learning models. These results provide strong support for the practical application and user satisfaction of multiperspective perception environmental facility design methods.

The proposed environmental facility design method for multiangle perception provides a new solution for practical environmental facility design by comprehensively utilizing joint fuzzy decision-making and transfer learning methods. In the future, we hope to further improve and optimize this method to better support practical environmental facility design work.

### Funding
The authors received no funding for this work.

### Competing Interests
The authors declare that they have no competing interests.

### Author Contributions
- Siconghui Yao conceived and designed the experiments, performed the experiments, analyzed the data, performed the computation work, prepared figures and/or tables, authored or reviewed drafts of the article, and approved the final draft.

### Data Availability
  Code of the proposed method and the original data.

## Supplemental Information

Supplemental information for this article can be found online at http://dx.doi.org/10.7717/peerj-cs.1855#supplemental-information.

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
