# Peer review of "Multiangle perception-oriented environmental facility design method based on joint fuzzy decision-making and transfer learning"

_PeerJ Computer Science, doi:10.7717/peerj-cs.1855_

## Round 0.1 · original submission · Major Revisions

In regards to this paper, one review is against it for publication, while two reviewers suggested a revision. I checked all comments from reviewers carefully, and I suggest it can get a revision chance for further consideration.

**Language Note:** The review process has identified that the English language must be improved. PeerJ can provide language editing services - please contact us at copyediting@peerj.com for pricing (be sure to provide your manuscript number and title). Alternatively, you should make your own arrangements to improve the language quality and provide details in your response letter. – PeerJ Staff

Reviewer 1 ·

Basic reporting

The paper presents an interesting premise focused on the development of a multi-perspective, perception-oriented environmental facility design method. The central idea of considering various factors, including user preferences, cultural backgrounds, and environmental conditions, is indeed relevant in the context of modern design needs. However, there are several critical issues in the paper that necessitate its rejection.

Experimental design

Very poor experimental design.
The paper does not have a single table to describe the results and compare them to previous works. The results that are discussed in the text are hard to follow.

Validity of the findings

Impossible to validate.
The paper provides bearly any details for the reproduction of the results.

Additional comments

The paper's introduction, while outlining the broad context and problem statement, suffers from a lack of clarity and organization. The information is presented in a somewhat disjointed manner, making it challenging to follow the logical flow of the research. The abstract should serve as a clear roadmap for the readers, outlining the objectives, methodology, and expected contributions, which is not achieved in this case.

The paper introduces two methods, joint fuzzy decision-making and Transfer learning, without providing sufficient detail about how these methods are combined and how they relate to the multi-perspective perception-oriented design. A comprehensive understanding of the proposed algorithm and the reasons behind selecting these methods is essential for the readers to evaluate the research's novelty and effectiveness.

The introduction references the limitations of existing research but fails to provide a meaningful comparative analysis. It would be beneficial to have a clear overview of what prior research has achieved and how the proposed joint algorithm improves upon these limitations. This would demonstrate the paper's contribution more effectively.


The paper exhibits issues related to language, grammar, and formatting that need correction. The language should be polished to ensure clarity, coherence, and readability. Proper formatting of the abstract is crucial to meet the standards of academic publications.

Finally, the figures of this paper are not up to the standard for a journal publication.

Reviewer 2 ·

Basic reporting

No comment

Experimental design

No comment

Validity of the findings

No comment

Additional comments

In this manuscript, author proposed a joint algorithm of perceptual fuzzy decision making and design Transfer learning methods are applied to the multi angle environmental facility design method. Author took into account the human, environmental, and cultural value parameters that affect the design results, and constructed a framework for joint processing. The experimental results showed that the joint model proposed in this manuscript can complete Transfer learning through continuous data collection under the condition of poor data quality in the early stage to improve its accuracy. Authors suggested to address the following comments and suggestions when preparing the revised version:
= Abstract: section needs to be re-drafted to be self-contained means it has to clearly show the hypothesis, methodology, techniques and tools used, and the results obtained.
= Keywords: Authors suggested to update the keywords by selecting more relevant terms. Keywords play important role in the appearance of the manuscript in scholars search which will give it more hits and more citations.
= What limitations author faced during this research work? If there is any.
= What assumptions authors made during the simulation phase of this research work? If there is any.
= Authors suggested to update the introduction and the related work sections by including more of the most recent published articles in the field.
= Conclusion: The conclusion should be abstracted so authors need to consider re-drafting it.
= Authors need to confirm that all acronyms are defined before being used for first time.
= Authors need to confirm that all mathematical notations are defined when being used for first time.
= Authors suggested to proofread the manuscript after addressing all comments to avoid any typo, grammatical, and lingual mistakes and errors.
= Authors advised to make sure that the format of all references is matching and complying with journal requirements and format.

Reviewer 3 ·

Basic reporting

1. In this paper, the Figures, tables and algorithms are placed at the end of the article, and I am not sure if this is intentional. I would suggest optimizing the typography to the relevant text to improve readability.
2. The introduction section should provide a clear background and motivation for the research. However, in this paper, the current introduction is too general and does not provide enough background information for the reader to understand the significance of the study. I suggest adding more details about the importance of environmental facility design and its specific challenges, as well as clarifying the research questions and objectives. It is also recommended to optimize the abstract section to highlight the work of this paper.
3. The literature review section should provide a comprehensive overview of the current state of knowledge in the field and explain how the proposed research builds on and adds to existing knowledge. However, in the current literature review, there are only five pieces of relevant literature and the analysis lacks depth. I recommend expanding the literature review to include more relevant studies and critically analyze the findings and limitations.

Experimental design

1. The paper only briefly mentions the research methodology used, but does not describe in detail the implementation steps, data collection process, and analysis methods. This prevents the reader from reproducing the study and raises questions about the reliability of the study. It is recommended to add relevant descriptions to clarify the details of the research methodology.

Validity of the findings

no comment

Additional comments

no comment

---

## Round 0.2 · Minor Revisions

As per comments from the reviewer, this work should undergo a minor revision to resolve the language problems

Reviewer 2 ·

Basic reporting

Authors revised and enhanced the manuscript according to reviewers comments and suggestions from previous review round. The quality standards of the manuscript meets with the journal ones. However, going through out the manuscript one can easily figure out some lingual and grammatical issues. So, authors advised to get the manuscript proofread by an English fluent speaker to avoid such issues. Also authors need to double check and confirm that all references are formatted according to journal references style and format.

Experimental design

no comments

Validity of the findings

no comments

Additional comments

no comments

Reviewer 3 ·

Basic reporting

no comment

Experimental design

no comment

Validity of the findings

no comment

---

## Round 0.3 · accepted · Accept

The current version can be accepted for publication.

Reviewer 2 ·

Basic reporting

Thanks to the authors for the review work done. The authors have appropriately addressed and corrected all the issues as per my previous comments. The related work has been enriched, and the indistinct description, as well as deficient analysis, have been improved and refined. More discussions have also been added. This paper has been comprehensively improved, in terms of correctness, completeness and readability to reach the standard for publication.

Experimental design

.

Validity of the findings

.

Additional comments

.

Reviewer 3 ·

Basic reporting

no comment

Experimental design

no comment

Validity of the findings

no comment